# Recent Advances in Biomolecular Patho-Mechanistic Pathways behind the Development and Progression of Diabetic Neuropathy

**DOI:** 10.3390/biomedicines12071390

**Published:** 2024-06-23

**Authors:** Yashumati Ratan, Aishwarya Rajput, Ashutosh Pareek, Aaushi Pareek, Ranjeet Kaur, Sonia Sonia, Rahul Kumar, Gurjit Singh

**Affiliations:** 1Department of Pharmacy, Banasthali Vidyapith, Banasthali 304022, Rajasthan, India; kailashaish@gmail.com (A.R.); ashu83aadi@gmail.com (A.P.); aayushipareek26@gmail.com (A.P.); 2Adesh Institute of Dental Sciences and Research, Bathinda 151101, Punjab, India; drranjeetkaur93@gmail.com; 3Department of Pharmaceutical Sciences, Guru Nanak Dev University, Amritsar 143005, Punjab, India; sonia01pharma@gmail.com; 4Baba Ragav Das Government Medical College, Gorakhpur 273013, Uttar Pradesh, India; dr.rahul707@gmail.com; 5Department of Biomedical Engineering, University of Illinois Chicago, Chicago, IL 60607, USA

**Keywords:** diabetic neuropathy, molecular mechanism, pathogenesis, polyol pathway, Schwann cells, neuropathic pain

## Abstract

Diabetic neuropathy (DN) is a neurodegenerative disorder that is primarily characterized by distal sensory loss, reduced mobility, and foot ulcers that may potentially lead to amputation. The multifaceted etiology of DN is linked to a range of inflammatory, vascular, metabolic, and other neurodegenerative factors. Chronic inflammation, endothelial dysfunction, and oxidative stress are the three basic biological changes that contribute to the development of DN. Although our understanding of the intricacies of DN has advanced significantly over the past decade, the distinctive mechanisms underlying the condition are still poorly understood, which may be the reason behind the lack of an effective treatment and cure for DN. The present study delivers a comprehensive understanding and highlights the potential role of the several pathways and molecular mechanisms underlying the etiopathogenesis of DN. Moreover, Schwann cells and satellite glial cells, as integral factors in the pathogenesis of DN, have been enlightened. This work will motivate allied research disciplines to gain a better understanding and analysis of the current state of the biomolecular mechanisms behind the pathogenesis of DN, which will be essential to effectively address every facet of DN, from prevention to treatment.

## 1. Introduction

Diabetic neuropathy (DN) is characterized by severe morbidity and pain, along with an impairment of sensory function that begins in the lower limbs. Fifty percent of diabetic patients gradually develop DN over time. It is characterized by a set of clinical or subclinical signs that are primarily caused by dysfunction in the peripheral nervous system (PNS) [1]. The etiopathogenesis of DN is multifaceted and mostly consists of changes in metabolic pathways (polyols, hexosamines, etc.) and vascular pathways (damage to endothelium) linked to neurodegenerative diseases and chronic inflammation [2]. The pathogenic mechanism states that in DN, the destruction of motor neurons occurs when sensory neurons’ distal terminals begin to first experience neuropathy [3]. Multiple factors contribute towards the development and progression of neuropathies, followed by diabetes mellitus (DM). DN has been found to be linked with hyperglycemia, raised blood triglyceride, and abnormal blood cholesterol. Elevated glucose in the blood affects the capillary walls, which transport oxygen as well as nutrients to nerves. Hyperglycemia in patients with DM can have detrimental consequences on the motor, autonomic, and sensory neurons of PNS [4,5].

Symmetric and asymmetric neuropathies require different diagnostic and treatment techniques; thus, it is critical to distinguish between them clinically. Diabetes patients typically have more than one type of neuropathy, even though distal symmetric polyneuropathy is the most prevalent type in both type 1 diabetes mellitus (T1DM) and type 2 diabetes mellitus (T2DM). It is imperative to bear in mind that neuropathy may not always result from diabetes in all patients. Diagnosing DN requires ruling out other etiologic variables in addition to diabetes [6]. The signs and symptoms of DN are dependent on its different types and the number of damaged nerves. For example, some patients with DN may have mild to moderately significant sensory loss on physical examination, although having minimal complaints. In contrast, individuals may experience a profound neuropathic impairment in the absence of any symptoms [7,8]. The most commonly reported symptoms are paresthesia (a sensation of pins and needles), numbness, sensory loss, and excruciating pain that keeps patients up at night. That pain could feel dull, achy, stinging, or searing [9].

A cohort study on the prevalence of neuropathy in DM revealed that 66% of the subjects suffered from T2DM and 59% suffered from T1DM [10]. About 10% of patients had DN at the point of diabetes diagnosis. Regretfully, within ten years of the diabetes’s progression, half of the patients are predicted to develop DN [11]. If appropriate care is not given, ulcers can lead to diabetic foot, one of the vascular problems associated with DN, and ultimately, to amputations [1,12]. Furthermore, neuropathy has been identified in 18.8–61.9% of patients with T2DM in India. Peripheral neuropathy is one among various kinds of neuropathies that impact more than 50% of people with diabetes worldwide [13].

In recent years, several classifications for DN have been proposed. However, there is not a classification that is acknowledged by everyone. The majority of writers usually agree with Thomas’ 1997 classification of neuropathy in diabetic individuals. According to Thomas [14], there are four primary categories of diabetic neuropathies: hyperglycemic, symmetric polyneuropathy, focal and multifocal neuropathy, and mixed forms. Diffuse symmetric, mononeuropathy, radiculopathy, or polyradiculopathy are the three primary categories into which the American Diabetes Association has recently divided DN [8]. Figure 1 displays the DN classification based on clinical presentation.

Glycemic control, which is attained by a balanced diet and lifestyle in addition to medication therapy, is essential for controlling DN. The type of DN and its symptoms determine the precise medication that is to be prescribed. Non-steroidal anti-inflammatory drugs, antidepressants (imipramine and amitriptyline), and antiepileptic medications (gabapentin and pregabalin) are commonly used to treat pain [15,16]. Alpha lipoic acid and vitamin B12 have been studied together in relation to the symptomatic therapy of DN in recent years [17]. Making lifestyle adjustments like giving up smoking, exercising, and having a healthy diet is also essential for managing diabetes-associated secondary problems [18].

There are not enough therapeutic options available for DN, regardless of the enormous expenses incurred to people and society. This is partly because the condition is complex and unanticipated, and since palliative care is the only treatment available, there is no comprehensive diagnosis plan or rigorous testing to support it. Although there is now no specified course of treatment, clinical trials for various DN interventions are being carried out [19]. The primary intent of the present study is to discuss and generalize the molecular mechanisms behind the etiopathogenesis of DN.

## 2. Risk Factors of DN

The risk factors for DN are either modifiable or non-modifiable, as outlined in Table 1. Age, the duration of diabetes, and height are the only risk factors for DN that cannot be modified or treated, while the majority of other risk factors are modifiable.

In diabetics, a number of endogenous elements secreted by the liver, skeletal muscle, endothelium, and various other tissues are known to play a crucial role in preventing damage and preserving the homeostasis of the vasculature. These endogenic variables associated with DM and its complications include insulin, nitric oxide (NO), platelet-derived growth factor (PDGF), and vascular endothelial growth factor (VEGF). An unbalance between injury and endogenous variables commences the onset of diabetic vascular complications [29,30], such as DN, diabetic retinopathy, diabetic nephropathy, and atherosclerosis. By counteracting the effects of oxidative stress, inflammation, and toxic advanced glycation end products (AGE), these endogenous factors serve a protective function [31]. DM covers two major mechanisms: increased blood sugar and impaired insulin signaling, either from insulinopenia (T1DM) or insulin resistance (T2DM). An aspect that contributes to the pathophysiology of DN is the interruption of direct insulin signaling, which has an effect on sensory neurons [32]. Reduced insulin receptor substrates, increased Ser/Thr phosphorylation state, increased Tyr Phosphatase action, decreased phosphoinositide-3-kinase (PI3K)/Akt kinase action, and defects in glucose transporter 4 (GLUT-4) activity are the catalytic consequences of altered insulin receptors that lead to insulin resistance. DM patients have also been observed to have the combined effect of elevated pro-inflammatory cytokines, saturated fatty acids, and chronic hyperinsulinemia conditions for these biomolecular abnormalities [33]. Basic proteins called PDGF are retained in the α granules of platelets. Scientific research has demonstrated that DM and related complications are influenced by the stimulation of the PDGF pathway. In T2DM, the PDGF pathway is abnormally active. It encourages atherosclerosis, diabetic wound repair, pancreatic β cell proliferation, and insulin resistance [34,35].

VEGF levels in human blood may be an indicator of how severe endothelial dysfunction is in diabetic individuals, which can result in DN and other microvascular consequences from the disease. The primary proangiogenic growth factor, VEGF, promotes endothelial cell activation in vitro and raises vascular permeability in vivo, contributing to the development of diabetic microvascular problems [36]. By raising oxidative stress and stimulating the synthesis of vasoconstrictor chemicals, hyperglycemia stimulates endothelial cells. This results in hypoxia, a strong inducer of VEGF-A release. Vascular permeability factors, often referred to as vasculogenesis and angiogenesis, are subsequently triggered by the release of VEGF-A. According to Badrah et al. [37], individuals with DN had no distinction in their serum concentrations of VEGF-A from those without DN; however, those suffering from subclinical DN had far greater levels than those with confirmed DN. This implied that individuals with T2DM may experience more severe DN as a result of reduced VEGF-A amounts. NO is necessary for the endothelium to perform its normal functions, which include blood pressure management, vascular tone modulation, homeostasis management, and smooth muscular cell proliferation [38]. Diabetic patients’ absence of NO synthase results in DN because it prevents the peripheral nerves from becoming vascularized. It has been shown that treating the ensuing NO deficit in human DN subjects can lessen their excruciating symptoms [39]. DM lowers the bioavailability of NO, which causes endothelial dysfunction. Even so, injury to tissues and oxidative stress are linked to hyperglycemia-induced excessive output of NO [40].

A progressive physiologic decline in several organs characterized by loss of functioning diminished physiological reserve, and heightened susceptibility to illness and death, is called frailty [41]. DM and frailty are age-related diseases that frequently coexist. Frailty and mortality have been found to be significantly correlated in diabetic patients [42]. A subgroup analysis carried out by Miao et al. revealed a possible correlation between frailty and DN [43]. Frailty affects how quickly diabetes mellitus progresses, how strictly it is controlled, and which therapies are to be chosen. Poorer outcomes, such as a higher death rate in frailty, are linked to DM. The chance to think about focused therapies to lessen disability and functional decline is presented by the early identification of frailty in older individuals with DM [44]. The musculoskeletal and metabolic systems may sustain damage over time due to chronic systemic inflammation and insulin resistance, which are linked to DM and its complications, eventually resulting in frailty [45].

## 3. Molecular Mechanisms and Pathophysiology Underlying the Pathogenesis of DN

The DN is a distinct neurodegenerative condition of the PNS, which predominantly impacts sensory, autonomic, and finally motor axons. Patients with diabetes may experience a range of neuropathic symptoms, with hyperglycemia being a major contributing factor in certain etiology. DN has a complex pathologic process, and its exact molecular mechanism is still unknown. The plausible biomolecular processes indulged in the etiopathology of DN include the polyol pathway [46], protein kinase C (PKC) pathway [47], hexosamine pathway [48], poly ADP-ribose polymerase (PARP) [49], Advanced glycation end-product (AGE) [50], oxidative stress [51], mitochondrial dysfunction [52], Wnt pathway [53], Hh pathway [54], mitogen-activated protein kinase (MAPK) signaling pathway [55], impaired insulin signaling, Glycogen synthase kinase 3 (GSK3) [56,57], Nuclear factor kappa B (NF-κB) [58], Cyclooxygenase (COX), Lectin-like oxidized low-density lipoprotein (LDL) receptor (LOX) [59], Interleukin (IL) [60], neurotrophic and cellular signaling, tumor necrosis factor (TNF)-α [61], autophagy, satellite glia cells, Schwann cells [62]. Different biochemical mechanisms associated with nerve and neurovascular injury have demonstrated that oxidative stress could modify the typical operation of metabolic mechanisms implicated in DN.

### 3.1. Metabolic Mechanisms

#### 3.1.1. Polyol Pathway

The Polyol path serves as one of the most crucial pathways for comprehending the mechanism of DN [63]. Throughout the past fifty years of history, there has been an expanding corpus of experimental and circumstantial findings suggesting that hyperglycemia increases the activity of the polyol pathway, hence exacerbating the symptoms of DN [64]. In this process, glucose is metabolized to sorbitol by the enzyme aldose reductase (AR) along with the co-enzyme nicotinamide adenine dinucleotide phosphate hydrogen (NADPH). Fructose is subsequently produced when sorbitol dehydrogenase Nicotinamide adenine dinucleotide (NAD^+^) is present, producing nicotinamide adenine dinucleotide hydrogen (NADH) [65]. Fructose, sorbitol, and NADH are produced by this process, which ultimately results in a negative redox balance between NADH and NAD^+^, sorbitol accumulation, abundant fructose, NADH synthesis, and overuse of NADPH. Additionally, the boost in the functioning of the polyol pathway reduces the NADPH required to replenish the antioxidant glutathione. As a result, there is inadequate glutathione, which raises the reactive oxygen species (ROS) level and causes oxidative stress in the nerves, as shown in Figure 2. Consequently, the events that occur in this route collectively participate in the development of DM and its related side effects, including retinopathy, nephropathy, and neuropathy [64].

Changes in the polyol pathway cause hyperosmolarity because they cause high levels of sorbitol to accumulate in nerve cells, which is followed by the release of 2-aminoethane sulfonic acid and a carbocyclic sugar like myoinositol [66]. Because of this, Myoinositol and taurine levels in the cell decrease. The normal function of the Na^+^/K^+^ ATPase pump, which correlates with the normal conduction velocity of nerves, is significantly influenced by myoinositol. Reduced myoinositol causes a shortage in adenosine triphosphate (ATP) synthesis, which in turn causes PKC functioning to gradually decline. These modifications thus cause oxidative stress. The activation of PARP may result from this oxidative stress, which may cause stimulation of certain death pathways and, finally, apoptosis [67]. Abundant sorbitol and fructose reduce myoinositol uptake in cells by inhibiting the sodium/myoinositol co-transporter expression. This leads to disruption of the phosphoinositide signaling system, which in turn decreases sodium/potassium ATPase function in neurons and restricts the transmembrane sodium pump activation [68]. Because of this, nerve interaction is impeded, and as long as this state continues, the membranes of nerves start to break down [69]. Furthermore, it has also been stated that the endothelium pro-inflammatory and prothrombic substances are produced in part by the AR. The onset of DN is significantly influenced by these pro-inflammatory mediators [70,71]. Immunocytochemical studies have shown that endothelial cells possess AR in the dorsal root ganglion (DRG), perivascular sympathetic axons, and blood arteries. These findings lend credence to the idea that neuropathy is a microvascular aftereffect of DM [72].

#### 3.1.2. PKC Pathway

PKC is a family of serine/threonine-related protein kinases that influence numerous signaling pathways connected to cell division, growth, and death. They are essential for numerous cellular processes [73]. Chronic hyperglycemia may raise the intermediary byproducts of glycolysis, which in turn may cause PKC activation and the generation of AGE [74]. An increase in diacylglycerol occurs when fructose-6-phosphate changes into glyceraldehyde-3-phosphate, which then changes into dihydroxyacetone phosphate [46]. In this instance, those suffering from diabetes experience an activation of the PKC pathway because of a hike in diacylglycerol levels. Oxidative stress is brought on by the production of ROS, which is stimulated by this elevated activity. Conversely, the MAPK ROS pathway is also triggered by the PKC pathway, and NF-κB is activated [75]. Translocation of NF-κB in the nucleus initiates gene transcription. This produces nitric oxide (NO) and inducible nitric oxide species (iNOS), which are important factors in the development of DN by promoting apoptosis and oxidative stress [76]. The PKC pathway in the pathogenesis of DN is depicted in Figure 2.

Research indicates that PKC activation reduces endothelial NO synthase m-RNA synthesis, leading to a rise in superoxide generation. Furthermore, it has been discovered that PKC function inhibits the production of NO, which may subsequently either increase the activity of endothelin-1 or decrease the angiotensin II functioning, leading to ischemia and vasoconstriction. Higher amounts of glucose in smooth muscle cells trigger the PKC pathway that further promotes the release of the vascular endothelial growth factor (VEGF), a permeability-inducing factor, and angiogenesis [47,77]. Additionally, PKC activity has been found to increase the production of type IV collagen, fibronectin, and transforming growth factor (TGF)-β. This is most likely due to disrupted NO synthesis, which in turn increases the microvascular matrix protein buildup. A fibrinolysis inhibitor called PAI-1 upregulation has been found to be associated with arterial occlusion and consequent death of neurons [47,78,79]. Numerous studies have shown that the synthesis of TNF-α and angiotensin IV regulated by PKC stimulates the transcription factor NF-κB [80,81]. Subsequently, this affects the gene expression linked to inflammation and the generation of various NADPH-dependent oxidases, all of which enhance the synthesis of ROS.

#### 3.1.3. Hexosamine Pathway

The hexosamine pathway is one of the metabolic pathways which contributes in the development of DN as shown in Figure 2 [46,48]. Following the polyol-pathway-mediated transformation of glucose to sorbitol and fructose, the leftover glucose is transformed into glucose-6-phosphate and then fructose-6-phosphate, which enters the hexosamine pathway [82]. Uridine diphosphate N-acetyl glucosamine (UDP-GlcNAc) is the end product of the hexosamine pathway, which is initiated by the enzyme glutamine-fructose-6-phosphate transaminase (GFPT)-1, which then transforms fructose-6-phosphate into glucosamine-6-phosphate (GlcN-6P) [83]. Hyperglycemia accelerates the development of DN by a number of mechanisms, such as elevated amounts of VEGF, TGF-α/β, and PAI-1 transcription. This can be avoided by blocking the process-related enzyme fructose-6-phosphate amidotransferase (GFAT), which is responsible for glucose conversion into glucosamine-glutamine [84].

Furthermore, consistent with the outcomes of Buse et al. [85], it was demonstrated that the GFAT and UDP-GlcNAc activity was markedly increased in the muscles of mice. Conversely, a decrease in the UDPGlcNAc level was discovered, and this enhancement of insulin sensitivity was shown in the muscles of the rats that had undergone prolonged calorie restriction. In clinical trials, muscle biopsies taken from insulin-resistant people revealed increased GFAT activity [86]. Conversely, obese patients with severe hyperglycemia and insulin resistance had significant recovery with insulin therapy, along with a 40% rise in UDPGlcNAc concentrations in their muscles. Even so, in contrast to non-diabetic controls, adipocytes from individuals with T2DM were linked to higher levels of circulating free fatty acids, leptin, and UDPGlcNAc [87].

#### 3.1.4. PARP

In a healthy body, PARP plays a role in apoptosis, mitochondrial (Mt) homeostasis, cellular proliferation, DNA repair, and inflammation. Conversely, hyperglycemia has been linked to DNA damage and PARP hyperactivation in Schwann cells [88]. Numerous research findings indicate that DNA single-strand breaks caused by ROS and free radicals are required for PARP activation [89,90,91]. Furthermore, excessive activation of PARP slows down glycolysis, which hinders the functioning of ATP and results in an energy shortage because NAD^+^ is used, and glyceraldehyde-3-phosphate dehydrogenase (GAPDH) is simultaneously inhibited. This leads to programmed necrosis [92]. Moreover, the oxidative-stress-induced destruction of Mt DNA impairs the regulation of energy needed for neuronal function, or the transmission between neurons, and harms nerve tissue and the DRG. It has been discovered that these damages are the main causes of pain perception in individuals with DM [93]. Thus, oxidative stress poses a risk to the functioning and health of cells, leading to tissue destruction. This lends credence to the theory that reducing the damage done to neurons and nerve tissues can reduce the pain that comes with DN [94].

#### 3.1.5. AGE

The AGE pathway is triggered by chronic hyperglycemia and leads to the breakdown of fructose-6-phosphate into glyceraldehyde-3-phosphate, as shown in Figure 2. The Maillard reaction, a complex series of events, is what causes the non-enzymatic glycation of lipids, proteins, and peptides with reducing sugars to produce AGE [95,96]. Essential proteins are cross-linked by AGE, which changes their functionality and damages cells. Additionally, AGE attaches to surface receptors of the cell, most notably the receptor for AGE (RAGE), which initiates a harmful downstream signaling cascade partly through NF-κB stimulation. AGE stimulation of RAGE has been associated with inflammatory processes, vasoconstriction, and lack of neurotrophic function in the rodent PNS; on the contrary, AGE accumulation has been observed in the peripheral nerves of people with DN and T2DM [97,98]. Aminoguanidine showed strong pre-clinical potential in preventing the generation of AGE in a mouse model of DM and DN; however, the drug’s negative effects prevented it from being developed as a therapy in humans [99,100]. Serum concentrations of sRAGE, a soluble extracellular domain of RAGE that blocks RAGE, are highly correlated in humans with diabetic nephropathy and cardiovascular disease but not yet with human DN [101,102]. To ascertain whether RAGE activation is a significant therapeutic target in DN, more thorough investigations akin to those conducted on diabetic nephropathy are proposed [103].

A study that involved the biopsy of 45 diabetes patients’ sural nerves revealed that the upregulation of RAGE in Schwann cells contributed to the pathogenesis of DN [104]. The results also showed that NF-κB, a crucial mechanism causing inflammatory neuropathies, was triggered by the activation of the RAGE pathway [105]. Clinically, activated NF-κB and RAGE were found co-localized in the vasa nervorum in sural nerve samples from DN patients [106]. Diabetic individuals exhibited fewer collateral vessels and elevated levels of endothelial RAGE in comparison to non-diabetic controls. These findings are interconnected with a higher risk of amputation of the lower limb. Consequently, because of its impact on the micro-vessels found inside sensory neurons, the AGE–RAGE–NF-κB axis may be in charge of the DN development [106].

#### 3.1.6. Oxidative Stress

Apart from the detrimental effects of hydroxyl radical, hydrogen peroxide, superoxide radical, and reactive nitrogen species, oxidative stress serves a crucial role in the pathogenesis of DN because of the imbalanced generated free radicals and free radical scavengers inside the cells [51,107]. Among patients with DM, hyperglycemia—a result of elevated blood glucose and reduced amounts of triglycerides and cholesterol—is a major factor influencing the generation of free radicals. This is brought on by the oxidation of glucose and the accumulation of ROS by non-enzymatic protein glycation. These can negatively impact cells and trigger apoptosis, which in turn causes neuropathy [108]. These ROS, or transitory reactive chemical substances, can damage cells and induce oxidative stress because they contain one or more unpaired electron pairs. Cellular injury is the consequence of damage to molecules and components of cells [109]. Furthermore, oxidative changes brought on by ROS like hydrogen peroxide and superoxide ions can result in the reduction in transcription factors like B-cell lymphoma 2 (Bcl-2), which are crucial for cellular survival and are in charge of regulating protein expression. However, research has shown that pro-apoptotic protein actions, including c-Jun N-terminal kinase, PARP, and Bcl-2 associated X protein (BAX), are upregulated in DN cases [110,111].

Decreased nutritious blood flow to the endoneurium, which may be caused by anomalies in motor and sensory nerve transmission velocities, has been shown in numerous investigations to cause sensory neuropathy and even small- and large-fiber degeneration in sensory nerves [112]. Additionally, enhanced oxidative stress and stimulated PARP have been shown in PNS, Schwann cells, and the vasa nervorum in a streptozotocin (STZ)-induced animal model of DN [113,114]. Furthermore, clinical studies showed that PARP activation and microvascular oxidative stress were clearly visible in the blood capillaries of individuals with DM [115,116].

#### 3.1.7. Mitochondrial Dysfunction

It has been indicated that the oxidative degradation of magnesium in DRG neurons, axons, and Schwann cells serves as a common pathway for DN [117,118,119,120,121,122]. A significant portion of diabetes’ long-term pathophysiology is brought on by ongoing hyperglycemia. ROS rises quickly at its highest point, and average Mt size grows by 50% in cultured prenatal rat neurons exposed to high glucose [123]. This is combined with the Mt membrane potential losing its control. Reduction in ROS production and DRG neuronal damage is achieved by inhibiting the adenosine nucleotide translocase and maintaining the Mt membrane potential. Findings that uncoupling proteins can stop inner membrane potential hyperpolarization and lower ROS production provide more proof that hyperglycemia destabilizes the inner Mt membrane potential [124]. If analogous alterations take place in vivo in DN, the outcomes still remain unknown. Reverse electron flow from Complex I and III of the respiratory chain may increase, and Mt electron carriers may be harmed by the rise in ROS. Long-term diabetes has been demonstrated to produce lowered respiratory chain operation, depolarization of the inner Mt membrane potential, and reduced Mt DNA levels in DRG neurons [121,125,126,127]. Impaired Mt function also leads to an issue with energy homeostasis. Neurons require a lot of energy to keep their membrane potential intact and have a big surface area [128]. Axonal transport systems, which also require energy, are necessary for the survival of distal axons, which may be located at a significant distance from the soma in humans. Overall, Mt dysfunction causes a cell to have a low intrinsic aerobic capacity, which, over time, may cause axonal and neuronal degeneration [129,130,131]. Therefore, in situations when there is a great deal of Mt degeneration, Mt regeneration might be protective.

### 3.2. Role of Other Signal Transduction Pathways/Enzymes/Processes in DN

#### 3.2.1. Wnt Pathway

The Wnts are a broad family of protein ligands that affect a number of functions, such as cell fate regulation, cell polarity formation, and embryonic initiation [132]. The canonical β-catenin-dependent and the non-canonical β-catenin-independent pathways compose the two pathways that make up the Wnt signaling pathway. Canonical pathway activators include Wnt1, Wnt2, Wnt3, Wnt3a, Wnt8a, Wnt8b, Wnt10a, and Wnt10b. Non-canonical pathway activators include Wnt4, Wnt5a, Wnt5b, Wnt6, Wnt7a, Wnt7b, and Wnt81 [133,134]. Wnt components are believed to promote the action of two kinases, Ca^2+^/calmodulin-dependent protein kinase II (CamKII) and PKC, mediated by G-protein coupling [135,136]. The Wnt signaling oversees preserving the equilibrium between the differentiation and proliferation of growing tissue. Multiple cells, including brain, mammary, and embryonic stem cells, are known to express Wnt proteins [137]. A research investigation was carried out to validate the role of the bone-regulating Wnt pathway in patients with Charcot arthropathy. The results showed that the levels of Wnt-1, Dickkopf-related protein-1 (Dkk-1), and sclerostin were lower in these patients than in diabetic controls [138]. Sclerostin amounts were significantly higher in the control group of diabetics compared to all other groups, while Wif-1 levels were significantly higher in the healthy controls. It has been established that increased spinal cord area index (SCAI) can prevent the progression of DN by preventing the Wnt/β-catenin upregulation. This indicated that treating DN by focusing on SCAI signaling might be an alternate therapeutic strategy [139]. The Wnt pathway in the pathogenesis of DN is illustrated in Figure 3.

#### 3.2.2. Hedgehog (Hh) Pathway

Inflammatory reactions, sensitivity to insulin, the breakdown of lipids, and complications associated with diabetes are all influenced by the Hh signaling system. Nearly every feature of the animal body plan is regulated by the proteins belonging to the Hh family, including cell development, their survival, and death [140,141]. In the nerve cells of STZ-induced diabetic rats, a decline in desert Hh (Dhh) expression has been observed by Calcutt et al. [142]. Lower motor nerve conduction velocity (MNCV) and sensory nerve conduction velocity (SNCV), decreased NBF, lowered pain inception when subjected to heat and/or formalin, and decreased levels of nerve growth factor (NGF) and neuropeptides are all linked to this downregulation of Dhh expression. According to reports, Hh protein is essential for ensuring the proper development of the peripheral nervous system [54]. Endoneuria permeability was induced, and capillary density was increased in Dhh-deficient mice. Smoothened conditional KO mice have demonstrated that this enhanced permeability is only caused by altered Hh signaling in senescent endothelial cells (ECs). Furthermore, NP and reduced responsiveness to painful stimuli can be induced in ECs by simple impairment of Hh signaling. Subsequently, it was found that the nerves of Leprdb/db T2DM mice lacked Dhh expression, which is thought to be correlated with enhanced endoneuria capillary permeability and declined levels of claudin 5 (Cldn5) [143]. The Hh pathway in the pathogenesis of DN is shown in Figure 3.

#### 3.2.3. MAPK Signaling Pathway

It has been revealed [144,145] that MAPKs are important players in the transmission of signals to various stimuli. Three MAPK families have been identified in mammalian cells. These include c-Jun N-terminal kinase (JNK), p38, and extracellular signal-related kinase (ERK) [73,145]. JNK and p38 cause neuronal apoptosis, while ERK1/2 is in charge of neural survival [146,147,148]. Nevertheless, it has been reported that ERKs contribute in the emergence of neuropathic pain [149,150]. Many investigations using STZ-diabetic rats have shown that the neurons in the spinal ganglia are overexpressed in ERK, p38, and JNK [151]. Individuals with T1DM and T2DM have been found to have higher sural nerve p38 and JNK levels [152,153]. In diabetic rats exhibiting DRG JNK activation, abnormal neurofilament phosphorylation can be seen [154]. Based on a different study, diabetic rats experience better neurological restoration because of the reduction in continuous JNK activation in DRG neurons. Moreover, the DRG of STZ-induced DM rats exhibited p38 activation [155].

#### 3.2.4. Impaired Insulin Signaling

Impaired insulin signaling in diabetes may be one of the primary factors that contribute towards PNS dysfunction and neuropathic symptoms [32]. It is also critical to keep in mind that DN is a multifaceted disease, meaning that decreased neuronal insulin signaling and hyperglycemic damage may be related. On the other hand, neuronal damage brought on by hyperglycemia injury is irreversible since insulin support is diminished. Inadequate PNS insulin support can lead to diminished neurotrophic properties, like neuron regeneration, poor functioning of Mt, altered neuropeptide/neurotransmitter modulation, and impaired glucose metabolism [156,157]. Insulin resistance has been shown by Hackett et al. [62] to be the cause of poor lipid metabolism in a variety of bodily cell types through the PI3K/AKT/mammalian target of rapamycin (mTOR) pathway. Developmental myelination and myelin maintenance in Schwann cells possess a role in insulin/Insulin-like growth factors (IGF1) signaling. The abnormal mTOR pathway is the cause of this mechanism.

#### 3.2.5. GSK3

The GSK3 protein kinase family helps in the phosphate addition to serine and threonine residues in amino acids. Mammals have been found to possess GSK3 isoforms, including GSK3α and GSK3β, which are encoded by different genes [158,159]. Key elements in multiple signaling pathways, including those governing cellular proliferation, glucose regulation, migration, and cell death, are GSK3α and GSK3β. They are also essential in controlling the rapid anterograde axon transport process. Numerous research works indicate the role of GSK3β in regulating inflammation throughout the PNS and central nervous system (CNS) [160,161,162]. GSK3 was shown to be reduced in experimental DN subjects by King et al. [156]. In vitro research revealed that compared to control rat tissues, the hippocampus and fresh sciatic nerves from rats with DM and control groups were considerably more phosphorylated on GSK3β and insulin receptors. In DN models of experimentation, a decrease in the level of neural phosphorylated GSK-3α/β (Ser21/9) has been recorded [163,164]. A further study found that both APP-transgenic mice and STZ-induced diabetic mice had significantly decreased levels of GSK3β in their sciatic nerves [165]. GSK-3 signaling in the pathogenesis of DN is depicted in Figure 3.

#### 3.2.6. NF-κB

A transcriptional factor called NF-κB is essential for immunological, inflammatory, and apoptotic responses. Stimuli that cause inflammation activate NF-κB. The role of NF-κB activation in DN pathogenesis has been depicted in Figure 4. Many investigations have shown that the kidney [166,167,168], sural nerves [58], monocytes [169], endothelial cells [169], and peripheral blood mononuclear cells all have elevated NF-κB expression [170]. Additionally, DRG, sural, and sciatic nerves in mice with DM showed higher NF-κB activation compared to normal control mice. Perineurium, epineural vessels, and endoneurium from the sural nerve biopsies of patients with diabetes and reduced tolerance to glucose have been found to have active NF-κB. Additionally, extracted Schwann cells cultivated in high glucose media showed increased NF-κB expression as compared to lower glucose. Besides contributing to other transcription factors, hyperglycemia and oxidative-stress-mediated NF-κB activation lead to the increased regulation of inflammatory genes, COX-2, endothelin-1, iNOS, and cell adhesion molecules [171,172]. Upregulation of the NF-κB p65 subunit has been observed during acute and chronic inflammatory demyelinating polyneuropathies, indicating a critical function for NF-κB in controlling the progression of inflammatory demyelination [173].

#### 3.2.7. COX and LOX

COX enzymes are considered to be the rate-limiting stage in the production of prostaglandin (PG) and are an essential component of arachidonic acid metabolism [174,175]. There are two primary kinds of COX: COX-1, which is involved in maintaining cellular homeostasis; and COX-2, which is produced less frequently under normal circumstances. A mechanism that has been linked to the pathophysiology of DN is the glucose-mediated modification of the COX pathway, which leads to the impairment of PG generation and functioning [176]. Growth factors, inflammatory cytokines, PKC activation, tumor promoters, and oxidative stress all play a role in COX-2 overexpression [177,178]. A number of pathologic diseases, including diabetes, are associated with the upregulation of COX-2. The overexpression of COX-2 is linked to the downstream inflammatory process activation and has specific implications for different tissues, as shown in Figure 4. Numerous investigations on diabetic rat models resulting from STZ revealed the elevated expression of COX-2 in neuronal cells [179,180]. COX-deficient mice are frequently shown to be immune to DM-induced deficits of nerve signaling, reduced blood flow surrounding myelin sheath, and intra-epidermal neural fiber volume, in contrast to wild-type COX-2(+/+) diabetic mice. These outcomes demonstrate the critical role COX-2 plays in nerve function.

LOX are primary enzymes engaged in the production of lipids, which include substances that are crucial in the initiation of diseases like diabetes, cancer, and cardiovascular disorders, as well as inflammation. The lipoxygenase family includes 12/15-LOX, a non-heme iron—with dioxygenase being one of its most significant members [181]. LOX is predominantly present in the macrophages, astrocytes, oligodendrocytes, smooth muscle cells, adipocytes, endothelial cells, and neurons of the nervous system [182,183]. It had been shown that the spinal cord, spinal ganglia, and sciatic nerves of mice given a high-fat diet had activated 12/15-LOX [184]. Investigations performed in vivo and in cell culture have indicated that elevated glucose levels can upregulate 12/15-LOX, which in turn can change the pro-inflammatory reaction, NF-κB, and metabolic and MAPK signaling cascades [185,186,187,188]. Obrosova et al. [189] showed that STZ-induced DM mice overexpressed 12/15-LOX in the sciatic nerves.

#### 3.2.8. Interleukin

IL are a class of cytokines that obtained their name from the way leukocytes communicate with one another. Over thirty different isoforms of IL have been found. IL-1, IL-6, and IL-8 are pro-inflammatory, while IL-4 and IL-10 are anti-inflammatory in their activities. Numerous neuropathic disorders have been linked to IL in research, including in both humans and animals. In comparison with normal controls, STZ-induced diabetes participants showed higher amounts of IL-6 mRNA in the sciatic nerve and DRG [187].

Bierhaus et al. [190] showed that sural nerve biopsies from DN patients had higher levels of IL-6 expression in contrast to controls. In T1DM patients, a direct correlation was found between elevated IL-6 expression and total cholesterol level, concentrations of fasting blood sugar, body mass index (BMI), and low-density lipoprotein cholesterol [191]. In contrast to patients with DM, DN patients had higher amounts of IL-17 and IL-13 in a single-blind controlled clinical investigation [192]. In STZ-diabetic rats, reductions in the cytokines IL-1α and IL-1β have been observed in the DRG and sciatic nerves [193]. STZ-induced DM rats showed an increase in pro-inflammatory cytokines, such as IL-1 and IL-6, in the spinal cord [194]. Furthermore, it has been discovered that diabetic individuals with foot ulcers and those exhibiting a high association with fibrinogen and C-reactive protein (CRP) overexpressed IL-6 [195,196].

#### 3.2.9. Neurotrophic and Cellular Signaling

The control of neuronal survival, growth, functioning, and plasticity is contingent upon neurotrophins [197,198,199]. Neurotrophins are known for activating two distinct kinds of receptors: a member of the TNF receptor superfamily, p75NTR, and the Trk family of receptor tyrosine kinases. Neurotrophins use these to trigger a variety of signaling pathways, such as those mediated by MAPK, PI-3K, and JNK cascades, as well as those mediated by RAS. They also control the production of proteins essential for regular neural activity, including ion channels and neurotransmitters, as well as considerations about cell fate, axon development, dendritic pruning, and neurotransmitter synthesis. Several other facets of brain activity are also regulated by these proteins [200]. Neurotrophin deficit is linked to the pathophysiology of DN [201]. NGF synthesis is inhibited in the epidermis of diabetic mice, and NGF supplementation exacerbates neuropathic alterations in small fibers and autonomic disease [202]. Elevated levels of the neurotrophic elements did not result in a discernible regeneration of the peripheral sensory fibers in a study that investigated the expression of NGF and other factors in the skin following kidney and pancreas transplantation in individuals with DN [203]. These results demonstrate the urgent need to understand more about the cellular components of DN. Techniques such as gene therapy and cell transplantation are now being researched to increase the efficiency of trophic factor production or delivery to the target tissues [204,205,206,207]. Additionally, there is proof that bone-marrow-derived cells, such as endothelial progenitor cells or mononuclear cells, can effectively treat a range of cardiovascular disorders due to their paracrine action. Because bone-marrow-derived cells express a wide range of angiogenic and neurotrophic cytokines, they are being utilized to treat DN and have demonstrated restoration of the signs and symptoms of the disease. Because of their capacity to promote cerebral neovascularization and their neuroprotective properties, endothelial progenitor cells are, in fact, the ones that exhibit the most promising therapeutic benefits [208,209,210,211].

#### 3.2.10. TNF-α

Monocytes and macrophages release TNF-α, an inflammatory cytokine that causes acute inflammatory processes [212]. It sets off a range of signaling cascades in cells that lead to either apoptosis or necrosis [213]. The inflammatory protein TNF-α is upregulated by a variety of mediators, including activated natural killer (NK) cells, macrophages, mast cells, clusters of differentiation 4 (CD4^+^) lymphocytes, and eosinophils. TNF-α’s primary function is to control immune cells. Both DN patients and those with diabetes had elevated amounts of TNF-α mRNA and plasma protein [214]. The role of TNF-α in DN has been verified in TNF-α(−/−) diabetic mice, which exhibited protection from SNCV and MNCV deficiency in contrast to normal mice. Based on various investigational studies, enhanced expressions of TNF-α and iNOS levels had an increased risk of developing DN [215,216]. In an in vitro research, treating rat microglia with more glucose boosted their mRNA expression of TNF-α and monocyte chemoattractant protein-1 (MCP-1), along with their production of these cytokines [217].

#### 3.2.11. Autophagy

By removing destroyed organelles and protein agglomerates, autophagy assists in shielding neurons from demyelination and bioenergetic crises [72,218]. Recent research has shown that T2DM patients have poor macro-autophagy. Li et al. [219] identified that in the hippocampal regions of diabetic mice with hyperglycemia, autophagy was decreased, whereas in mice without hyperglycemia, it remained unaltered. Elevated hyperglycemia directly inhibits autophagy in the primary hippocampus neurons of mice. Furthermore, autophagy provides protection against neurotoxicity caused by excessive hyperglycemia. Additionally, as reported by Li et al., GFP-LC3 revealed that elevated glucose inhibited autophagic flow via compromising autophagosome production. The generation of NO under conditions characterized by elevated glucose was the cause of the decline in autophagy. Ultimately, it can be stated that neurotoxicity in response to hyperglycemia is caused by autophagy impairment mediated by S-nitrosation of autophagy-related 4B (ATG4B). Mohseni et al. [220] examined the sural nerve physiological processes and morphological concepts at baseline and 11 years after normal glucose tolerance (NGT) and type 2 diabetes to see if there was a relationship between the autophagic process and sural nerve fiber pathology. It is interesting to note that at baseline, individuals with T2DM had reduced nerve amplitude than those with NGT. Hence, T2DM may have an impact on autophagy, which occurs in sural nerves. Since there is no FDA-approved treatment for DN at present, a great deal of research has gone into creating various therapeutic agents.

Sir Tuin 1 (SirT1) is a crucial metabolic sensor. It reacts to variations in the NAD^+^/NADH ratio within cells. It was suggested that diabetic memory-induced epigenetic alterations could be reversed by SirT1-aided FOXO3a signaling [221]. The pathophysiology of insulin resistance and vascular problems linked to T2DM may therefore be partially explained by SirT1 inactivation. In 2017, Yerra et al. [222] examined the impact of isoliquiritigenin (ILQ)-induced SirT1 activation on AMP kinase and peroxisome proliferator-activated receptor gamma coactivator 1-alpha (PGC-1α) signaling in the peripheral nerves of rats with DM and in neuro2a (N2A) cells subjected to elevated glucose (30 mM). The antidiabetic potential of the SirT1 activator and the well-known antioxidant ILQ have recently been investigated. On exposure to excessive glucose, both N2A cells and diabetic rats displayed decreased SIRT1 expression along with decreased autophagy and biosynthesis of mitochondria. When ILQ was administered, high glucose-exposed N2A cells and diabetic rats exhibited notable SIRT1 activation along with an upsurge in autophagy and Mt biogenesis.

#### 3.2.12. Malabsorption and DN

Because of abnormalities in the pancreatic exocrine and endocrine systems, DM is an endocrine condition [223]. Exocrine pancreatic insufficiency (EPI) is mostly caused by inadequate biochemical production from the pancreatic acinar cells. The endocrine and exocrine pancreas are both closely linked organs. In individuals with T2DM, EPI may result in a reduction in weight, steatorrhea, fat malabsorption, and gastrointestinal distress [224,225]. Notably, research has revealed that patients with DM and EPI are at a higher risk of morbidity and death due to problems resulting from malnourishment, as well as autonomic and peripheral DN [226,227]. The majority of young adult males with juvenile-onset DM exacerbated by DN experience diabetic diarrhoea and steatorrhea. Although widespread wasting and loss of other nutrients are extremely unusual, severe malabsorption of fat can take place. In patients with DM and DN, the mechanism of malabsorption may be linked to exocrine pancreatic dysfunction [228]. EPI results from a rise in proinflammatory cytokines such as interleukin (IL)-6, βIL-1, and tumor necrosis factor-alpha, which causes inflammation within the pancreas in people with T2DM. Patients with chronic heart failure also experience appetite loss as a result, which raises the possibility that EPI plays a part in cardiac autonomic neuropathy, particularly in those with T2DM [229]. Additionally, EPI ultimately results in fat malabsorption in diabetic patients due to poor entero-pancreatic reflexes caused by visceral neuropathy, fibrosis, pancreatic atrophy, and reduced acinar cells [230,231].

When autoimmune impairment and insulin shortage are absent, autonomic neuropathy and microvascular injury in those with T2DM may contribute to the development of EPI. Peptides and neurotransmitters from the pancreas, the stomach, and vagal stimulation modulate entero-pancreatic reflexes, which in turn mediate the exocrine pancreatic reaction that follows a meal consumption physiologically. Consequently, autonomic neuropathy’s interference with entero-pancreatic reflexes may compromise the exocrine functioning of the pancreas [232]. When fecal fat concentrations exceed 7 g per day, it may indicate EPI-linked fat malabsorption, which might worsen the quality of life for people with erratic, long-term DM [233]. Individuals with DM may experience DN that impairs sensory innervation as well as the regulation of intestinal and/or stomach impulses. Stomach contraction is impacted by excitatory pathway dysfunction, which can lead to prolonged gastric emptying and retained food vomiting. On the other hand, alterations to the inhibitory neural pathways lead to a decrease in the stomach’s ability to relax after eating. DN can cause abnormalities in the regulation of gastrointestinal movement, resulting in a range of symptoms such as constipation, diarrhoea, intestinal tension, and pain in the abdomen. DN can also impact the gut’s sensory nerves, and contingent upon the pathways affected, perception may be enhanced or diminished [234].

## 4. Glial Cells in the Pathogenesis of DN

### 4.1. Satellite Glia Cells (SGC)

The sensory and autonomic ganglia of PNS are home to satellite glial cells, which encircle every single neural soma in a narrow, taut sheath. The SGC become activated in response to a broad range of neuronal stress conditions. Animal studies of traumatic nerve injury [235], DM/DN [236,237,238,239,240], inflammation [241], chemotherapy [242,243], and herpes simplex infection [244,245] were among the conditions in which this has been observed. All of these outcomes point to the possibility that activation of SGC is a normal physiologic reaction to neural stress. It is noteworthy that SGC purinoceptor (P2X) 7 receptor activation mediates SGC-to-neuron signaling, which causes SGC to release ATP and TNFα [246] and increases the hyperexcitability of the encased neurons [247,248,249]. Moreover, emerging research indicates that neural stress may propagate across a ganglion by showing how SGC surrounding nearby sensory neurons react to nerve damage by causing a coupling of neural activity [248,250]. These findings are supported by research that demonstrates how mechanical hyperalgesia and DRG neuron hyperactivity are reduced when gap junctions are blocked [251]. SGC involvement in the pathogenesis of DN has been depicted in Figure 5.

In a T1DM model of rodents, Hanani et al. [252] reported that SGC activation is indicated by a five-fold spike in neuronal cells surrounding glial fibrillary acidic protein (GFAP)-positive SGC in the DRG of mice and a four-fold rise in neurons in the rat DRG. Based on GFAP expression in the DRG, it has been discovered that blocking P2X7 receptor overexpression in a diabetic rat model prevented SGC stimulation [237]. Furthermore, in diabetic rats, blocking the overexpression of P2X7 receptors decreased TNFα production, which in turn inhibited DRG neuron excitability and decreased hyperalgesia. Collectively, these findings imply that SGC is responsive to the neuronal stress caused by diabetes and may, hence, be involved in the development of DN. Alternatively, it is possible that elevated glucose directly affects SGC, causing SGC responsiveness in the absence of prior neural stress. The finding that AR is expressed by SGC but not by DRG neurons of both healthy and diabetic rats lends credence to this theory [253].

### 4.2. Schwann Cells

The majority of glial cells in the PNS are Schwann cells, which can be either myelinating or non-myelinating, and they encase every axon in a peripheral nerve. Schwann cells have the ability to modify neuronal biology and are more than just passive insulators of axons [254,255,256,257]. Disease conditions like diabetes may impair nerve homeostasis, glial-axon interactions, and Schwann cell functioning, which can ultimately result in fiber loss, neurodegeneration, and pain. Unspecified modifications in morphology in human nerve preparations, as well as in disease models in cats and rodents, clearly show signs of Schwann cell stress and reduced Schwann cell functioning [258,259,260]. As the disease progresses, slight segmented axonal demyelination and remyelination take place in the company of a normal axon [261,262], indicating that Schwannopathy may very well represent the initial stage in the pathophysiology of DN and the underlying primary injury to nerve fibers.

#### Schwann Cells and AR in DN

AR has been demonstrated in a number of studies for myelinating the Schwann cells of the somatic neurons. Schwann cell malfunctioning has been identified in various pre-clinical animal models of DM. These include the reduced expression of myelin-related proteins, and trophic factors originated by Schwann cells, such as ciliary neurotrophic factor (CNTF) 19 and desert hedgehog, which serve as crucial regulators of determining cell fate and are primarily indicated in peripheral nerves which cause the myelination of Schwann cells. By blocking AR, these modifications that could impact vascular and neuronal activity, as well as CNTF expression alterations, can be prevented [143]. According to the findings of additional studies conducted on cultivated Schwann cells, these cells are pushed toward an immature phenotype by an upsurge in the flow along the polyol pathway. Furthermore, mice that were subjected to galactose poisoning showed elevated polyol activity and acquired DN, which mirrored the pathogenesis of DN in humans. AR inhibition may help prevent DN [263]. Schwann cells’ involvement in the pathogenesis of DN is depicted in Figure 6.

Changes in Bcl-2, BAX, and caspase-9 expression in vitro have been linked to elevated-glucose-aided death in Schwann cells [264], suggesting Mt internal stress. There are several ways in which Mt failure has been linked to the pathophysiology of DN. For example, evidence derived from primary sensory neurons has linked mitochondria dysfunction to disturbed calcium homeostasis [265,266]. More recently, research has shown how deficiencies in the mitochondria in Schwann cells appear to trigger an unfavorable stress response that leads to the buildup of acylcarnitine and, upon release, axonal degeneration [267,268]. Elevated expression of ATP synthase subunits α and β and modification of the Schwann cell Mt proteome have also been linked to hyperglycemia. Additional effects of elevated glucose include reduced respiratory ability, which is brought on by a higher amount of oxygen consumption overall, indicating inadequate oxidative phosphorylation in Schwann cells [269]. Thus, by minimizing demyelination and boosting axonal survival, targeting Schwann cell mitotoxicity may provide novel insights into DN treatments.

## 5. Conclusions

About 30 to 90 percent of diabetic people worldwide suffer from DN, which is a complex and potentially fatal neurodegenerative disorder. DN is a collection of syndromic complications rather than a single disease. The long-term pathogenesis of DN is influenced by three main and closely connected pathophysiological conditions: oxidative stress, endothelial dysfunction, and chronic inflammation. Since DN abruptly affects both the CNS and PNS, nervous tissue suffers significant atrophy. Numerous studies demonstrate the severity of the condition and the various molecular mechanisms that interact to promote DN. A specific molecular mechanism for the development of DN after diabetes is not presented in any of the findings. The present study explores the potential relationship between a number of molecular and metabolic cascades, including polyol pathways, dyslipidemia, and glycolysis, which can cause microvascular damage and mitochondrial dysfunction in DN. Moreover, a comprehensive understanding of different signaling cascades and pathways including polyol, PKC, hexosamine pathway, PARP, AGE, oxidative stress, mitochondrial dysfunction, Wnt, Hh pathway, MAPK, impaired insulin signaling, GSK3, NF-κB, COX, LOX, IL, neurotrophic and cellular signaling, TNF-α, autophagy, satellite glia cells, and Schwann cells that may be responsible for the etiopathogenesis of DN have been delivered. Since, at present, there is no cure for DN, research into discovering medications that both suppress the process of disease development and activate the pathways important for preventing diabetic neurodegeneration is essential. From a therapeutic standpoint, this review highlights important research gaps that need to be filled towards an attempt to develop a promising drug candidate and its molecular target for DN. In a nutshell, DN research has evolved from research focused on distinct dysregulated pathways, such as AGE, PKC, and AR, into an entirely novel discipline that focuses on insulin sensitivity and resistance, understanding global whole-nerve metabolism, nutrient overload, and the glial cells as an integral factor in DN. Rapid advancements in comprehension of the involvement of PNS and CNS in the pathophysiology of pain in DN suggest new cross-talks between the various nervous system elements. Whilst much remains to be discovered, the latest developments in DN research offer novel insights that will probably lead to the development of much-needed mechanism-based therapeutics.

## Figures and Tables

**Figure 1 biomedicines-12-01390-f001:**
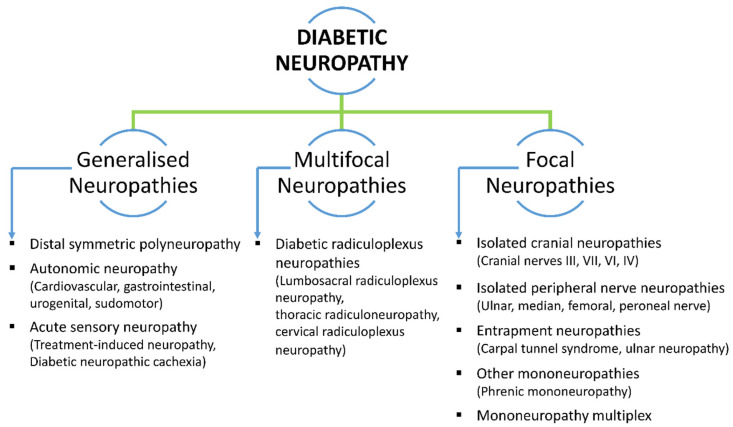
An illustration of the classification of DN based on clinical presentation.

**Figure 2 biomedicines-12-01390-f002:**
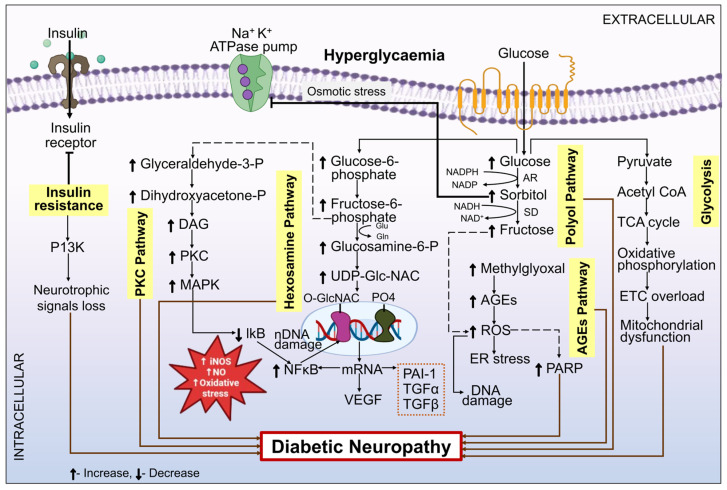
Schematic representation of the PKC, hexosamine, polyol, AGE, PARP, and glycolysis pathways’ contribution to the pathogenesis of DN. During hyperglycemic conditions, the breakdown of glucose in nerve cells saturates the glycolytic pathway, which simultaneously increases AR activity. In the polyol pathway, too much glucose is metabolized by AR and sorbitol dehydrogenase into fructose and sorbitol, respectively. This creates osmotic stress and ROS, which in turn damages DNA and eventually results in DN. Moreover, sorbitol prevents the synthesis of Na^+^/K^+^ ATPase, and glycolysis leads to the malfunctioning of mitochondria. Insulin resistance causes phosphoinositide-3-kinase (PI3k) to be inhibited and Akt to be upregulated, which results in the loss of neurotrophic signaling, which further contributes to DN. Fructose-6-phosphate, which is produced when glucose is metabolized, encounters the hexosamine pathway and increases the synthesis of Uridine diphosphate *N*-acetyl glucosamine (UDP-GlcNAC). This, in turn, causes hyperactivations of transforming growth factor (TGF)-β, Vascular endothelial growth factor (VEGF), Nuclear factor kappa B (NF-κB), and plasminogen activator inhibitor 1 (PAI-1), which in response promotes DN. Fructose-6-phosphate is transformed into glyceraldehyde-3-phosphate via the PKC pathway, which hyperactivates PKC and increases mitogen-activated protein kinase (MAPK) synthesis. This leads to NF-κB activation, which destroys nuclear DNA and results in neuropathy. Glyceraldehyde-3-phosphate is transformed into methyglyoxal via the AGE pathway, which causes disruption of energy control and a decrease in adenosine triphosphate (ATP) synthesis. This results in AGE upregulation and PARP hyperactivation. These subsequently trigger an upsurge in endoplasmic reticulum (ER) stress, which starts the DN pathogenesis.

**Figure 3 biomedicines-12-01390-f003:**
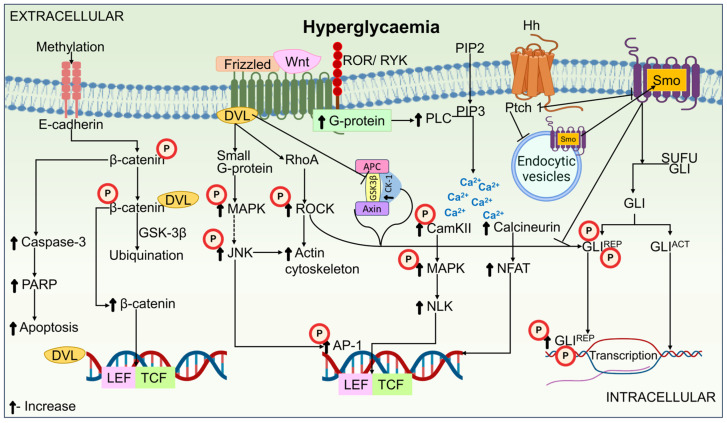
A schematic representation of Wnt, Hh, and GSK3 pathways in the pathogenesis of DN. β-catenin stabilizes and translocates to the nucleus to initiate Wnt signaling. The Frizzled receptor is bound by the Wnt ligand, which then draws the Axin degradation complex away from β-catenin. Following this, β-catenin and the transcription factors T-cell-specific factor (TCF) and lymphoid enhancer-binding factor (LEF) are able to move to the nucleus. Dephosphorylation and nuclear translocation of β-catenin are inhibited by the destruction of the axin degradation complex. Further, β-catenin is ubiquitinated by GSK-3β. A different way to start Wnt signaling is via activating the G-protein receptor through interactions with the receptor-like tyrosine kinase (RYK) and Retinoic acid receptor-related orphan receptor (ROR). In response to G-protein activation, the endoplasmic reticulum releases Ca^2+^, which in turn stimulates the nuclear activation of Nemo-Like Kinase (NLK) and the nuclear factor of activated T-Cells (NFAT). This process further stimulates Calcium/calmodulin-dependent protein kinase II (CamKII) and calcineurin. JNK nuclear activation is mediated by the small G-protein. Smothened (Smo) is inhibited when Hh binding activates patched 1 (Ptch 1). To produce the primary activated form of Glioma-associated Protein (GLI), Smo controls the disintegration of the suppressor of fused (SUFU)-GLI. Smo stimulates the production of activated GLI (GLIACT) but inhibits the GSK3/CK1α/PKA-mediated phosphorylation of repressed GLI (GLIREP). Target gene transcription is started by GLIACT translocation into the nucleus, whereas transcription is stopped by phospho-GLIREP.

**Figure 4 biomedicines-12-01390-f004:**
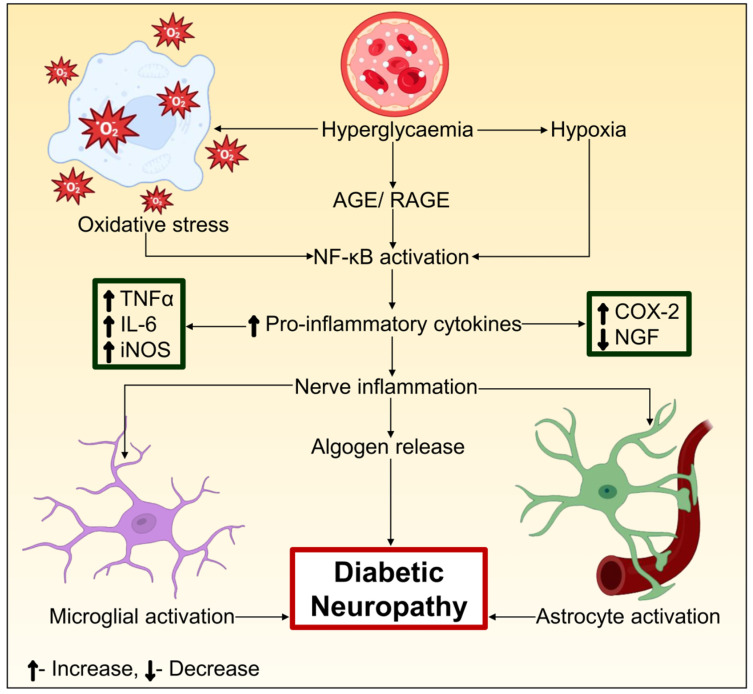
A schematic diagram of the role of inflammatory responses and NF-κB in DN. Slowdowns in the rate of nerve conduction are caused by ATPase activity, Na^+^, K^+^, and sorbitol in the nerves. The pathophysiology of DN is brought on by NF-κB activation, which also damages neuronal DNA. Leukocyte infiltration and a reduction in NGF in nerve cells are caused by NF-κB overexpression brought on by elevated ROS, PKC, and AGE. The activation of astrocytes and microglia is caused by the elevated expression of proinflammatory cytokines, including Interleukin (IL)-1b, IL-6, and TNF-α, which further contributes to the pathophysiology of DN. The activation of NF-κB leads to a rise in the quantity of COX-2 in nerve cells, which further activates the arachidonic acid pathway. Increased oxidative stress causes nerve damage by activating stress kinase MAPK.

**Figure 5 biomedicines-12-01390-f005:**
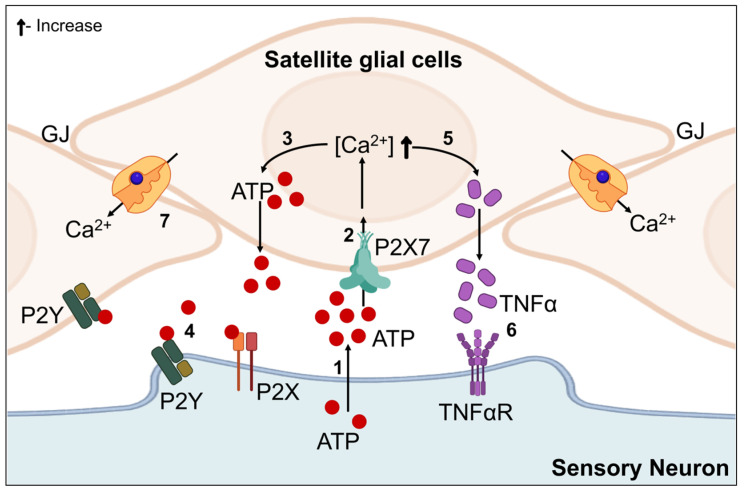
An illustration of Satellite glial cells in the pathogenesis of DN. ATP escapes into the extracellular space through the soma during neuronal stress. [Ca^2+^] rises as a result of secreted ATP activating purinergic receptors on the SGC, such as P2X7. An increase in [Ca^2+^] causes SGC to release more ATP, which raises the extracellular ATP level and activates P2X and P2Y receptors on neurons. Moreover, P2X7 receptor activation promotes the SGC discharge of cytokines like TNF-α, which links to and stimulates neuronal TNF-α receptors. Connexin channel overexpression results in a rise in gap junctions (GJ) and the intercellular coupling of SGC by Ca^2+^ waves, which are characteristics of activated SGC.

**Figure 6 biomedicines-12-01390-f006:**
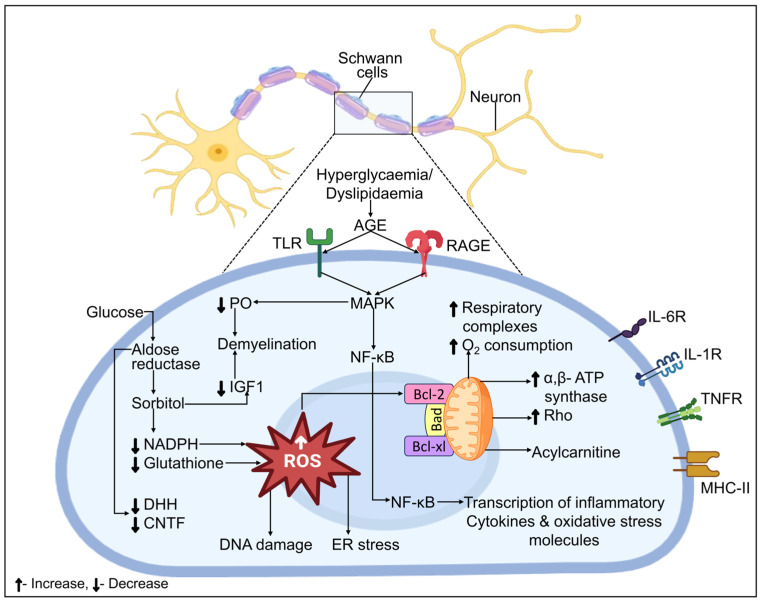
Schematic illustration of Schwann cell dysfunction and neurodegeneration in DN. AR shifts increase glucose concentrations into the polyol pathway, which reduces glutathione regeneration by lowering cytosolic NADH. Thereby, glucose metabolism results in damaging DNA, ER stress, decreased neurotrophic factor synthesis, and localized oxidative stress, all of which are responsible for the immature phenotype of the cells. The enhanced production of ATP synthase and respiratory chain complexes in Schwann cells, which increases the intake of oxygen and apoptotic signaling events, is another indication of mitochondrial stress brought on by hyperglycemia. Consequently, stressed Schwann cells emit acylcarnitine, which causes the degeneration of axons and fiber loss. RAGE/TLR stimulation by AGE leads to downstream signaling processes, which are partially mediated by NF-κB activation and MAPK, leading to the production of oxidative stress molecules and inflammatory cytokines. Schwann cells, which express MHC class II and a number of cytokine receptors, including TNFR, IL-1R, and IL-6R, have been identified as immune-competent cells.

**Table 1 biomedicines-12-01390-t001:** Risk factors associated with DN. Unmodifiable and modifiable risk factors linked with the development and progression of DN have been generalized in the table.

Risk Factor	Description	Reference
Unmodifiable risk factors associated with DN		
Age and duration of diabetes	It is commonly known that the duration of diabetes and age both raise the chance of acquiring neuropathy. DN is more common in older persons (>50 years) because hyperglycemia takes longer to cause nerve damage. All individuals with T2DM, along with those who have suffered from T1DM for longer than five years, should be suspected of having DN.	[20]
Height	In patients with diabetes, height has been shown to be an independent indicator of neuropathy and is thought to serve as a measure of neuronal length.	[21]
Modifiable risk factors associated with DN		
Hyperglycemia	In DN, hyperglycemia is the main risk factor. In T1DM, better glycemic management can stop DN from progressing, but in T2DM, it cannot stop distal polyneuropathy from occurring. Tight glucose management is beneficial because it slows the progression of DN in diabetic individuals, as clinical studies have demonstrated.	[22,23]
Obesity	After diabetes, obesity is the second most important metabolic risk variable for neuropathy. Furthermore, neuropathy has been reported to be more common in obese normoglycemic persons compared to lean controls in clinical studies, indicating that obesity is a separate risk factor for neuropathy.	[24,25]
Cardiovascular risks	Independent of glycemic management, cardiovascular risk factors such as low high-density lipoprotein, hypertriglyceridemia, abdominal obesity, dyslipidemia, and hypertension can raise the risk of DN.	[26]
Hypertension	The development of diabetes or the acceleration of chronic renal disease can lead to hypertension. Patients with diabetes may develop essential hypertension as a concurrent condition, or DN may result in secondary hypertension. Since blood pressure regulation is clinically addressed for the prevention of distal symmetric polyneuropathy, hypertension can exacerbate poly-neuropathies in diabetes patients.	[27,28]
Smoking, alcohol	Alcohol intake and smoking are also regarded as separate risk factors for DN.	[4,5]

## Data Availability

Not applicable.

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
