# Peer review of "Recent Advances in Biomolecular Patho-Mechanistic Pathways behind the Development and Progression of Diabetic Neuropathy"

_biomedicines, 2024, doi:10.3390/biomedicines12071390_

Round 1

Reviewer 1 Report

Comments and Suggestions for Authors

Recent Advances in Biomolecular Patho-mechanistic Pathways Behind the Development and Progression of Diabetic Neuropathy

Diabetic neuropathy is one of the most debilitating outcomes of diabetes mellitus and may cause pain, decreased motility, and even amputation. Hyperglycemia, apart from inducing oxidative stress in neurons, also leads to activation of multiple biochemical pathways which constitute the major source of damage and are potential therapeutic targets in diabetic neuropathy. A meta-analysis would be indicated, the mechanisms presented being also found in other articles. References must be up-dated.

Author Response

S. No.

Comments

Responses

REVIEWER 1

1

Diabetic neuropathy is one of the most debilitating outcomes of diabetes mellitus and may cause pain, decreased motility, and even amputation. Hyperglycemia, apart from inducing oxidative stress in neurons, also leads to activation of multiple biochemical pathways which constitute the major source of damage and are potential therapeutic targets in diabetic neuropathy. A meta-analysis would be indicated, the mechanisms presented being also found in other articles.

Thank you for your valuable feedback.

The present study is a narrative systematic review focusing on the various molecular mechanisms underlying the pathogenesis of diabetic neuropathy. We have thoroughly reviewed recent studies, including articles, review articles, meta-analyses, and clinical studies, that highlight the pathological molecular mechanisms and pathways involved in diabetic neuropathy. Our manuscript discusses these aspects to provide a comprehensive understanding of the interconnected pathogenic factors contributing to diabetic neuropathy.

While a meta-analysis could offer quantitative insights, our narrative approach aims to capture the breadth and complexity of the multiple mechanisms at play. This format allows us to explore and present the diverse and intricate molecular interactions and pathways that contribute to the development and progression of diabetic neuropathy. We believe this comprehensive overview is crucial for identifying potential therapeutic targets and understanding the disease's multifaceted nature.

2

References must be up-dated.

We have updated the references throughout the manuscript to include the most recent and relevant studies available. This ensures that our review reflects the current state of research in the field and incorporates the latest findings and advancements related to the pathogenesis of diabetic neuropathy.

Reviewer 2 Report

Comments and Suggestions for Authors

Dear Author's

Thank you for the opportunity to read the results of Your article. An interesting, informative article. A large amount of cited literature. Overall, I have no critical comments. I wonder why the authors did not attempt a systematic review, such a review always has greater scientific value - please explain. I have no other comments.

best regards for all Author's

Author Response

Responses to Reviewer 2 Comments

S. No.

Comments

Responses

REVIEWER 2

1

Dear Author's

Thank you for the opportunity to read the results of Your article. An interesting, informative article. A large amount of cited literature. Overall, I have no critical comments. I wonder why the authors did not attempt a systematic review, such a review always has greater scientific value - please explain. I have no other comments.

Best regards for all Author's

Thank you for your valuable feedback and positive comments on the structured presentation and clarity of the content.

The present study is a narrative systematic review focusing on the various molecular mechanisms underlying the pathogenesis of diabetic neuropathy. We chose this approach to provide a comprehensive overview of the diverse and intricate molecular interactions and pathways involved, which might be less constrained by the strict inclusion and exclusion criteria of a traditional systematic review.

However, we greatly appreciate your suggestion and will consider conducting a systematic review in future studies to further enhance the scientific value of our work.

Reviewer 3 Report

Comments and Suggestions for Authors

Recent developments in the biomolecular pathomechanistic pathways underlying the onset and progression of diabetic neuropathy are studied by a review.
Diabetic neuropathy affects patients with diabetes worldwideand could be very dangerous. Authors investigate the possible connections among several molecular and metabolic pathways that can lead to microvascular damage and mitochondrial dysfunction in diabetic neuropathy, such as the polyol pathway, dyslipidemia, and glycolysis. The writing is captivating and well-written. I suggest that the authors take into account endogenic variables associated with diabetes that may increase the risk of not just diabetic neuropathy but also other fatal disorders (10.1186/s12933-019-0885-2; 10.1186/s12877-021-02304-9).
Moreover, malabsorpion and diabetic neuropathy have been linked in the literature (10.1111/j.1464-5491.1992.tb01758.x; 10.1097/MCG.0b013e318159c654) and I suggest authors to consider this topic in discussion.

Author Response

Responses to Reviewer 3 Comments

S. No.

Comments

Responses

REVIEWER 3

1

Recent developments in the biomolecular pathomechanistic pathways underlying the onset and progression of diabetic neuropathy are studied by a review.

Diabetic neuropathy affects patients with diabetes worldwide and could be very dangerous. Authors investigate the possible connections among several molecular and metabolic pathways that can lead to microvascular damage and mitochondrial dysfunction in diabetic neuropathy, such as the polyol pathway, dyslipidemia, and glycolysis. The writing is captivating and well-written.

I suggest that the authors take into account endogenic variables associated with diabetes that may increase the risk of not just diabetic neuropathy but also other fatal disorders (10.1186/s12933-019-0885-2; 10.1186/s12877-021-02304-9).

Thank you for your valuable feedback and positive comments.

In response to your suggestion, we have included a discussion on endogenic factors associated with diabetes that may increase the risk of diabetic neuropathy and other complications. These additions can be found in the manuscript on lines 106-161.

If there are any additional aspects or specific endogenic factors you believe should be further elaborated, please let us know.

2

Moreover, malabsorption and diabetic neuropathy have been linked in the literature (10.1111/j.1464-5491.1992.tb01758.x; 10.1097/MCG.0b013e318159c654) and I suggest authors to consider this topic in discussion.

In response to your recommendation, we have included a discussion on the link between malabsorption and diabetic neuropathy in the manuscript (lines 649-685). We have also referenced the suggested articles to provide a comprehensive overview of this important aspect.

Please let us know if there are any additional details or specific points you believe should be further elaborated.

Round 2

Reviewer 1 Report

Comments and Suggestions for Authors

Recent Advances in Biomolecular Patho-mechanistic Pathways Behind the Development and Progression of Diabetic Neuropathy

Revised